# A Semi-Automatic Coupling Geophone for Tunnel Seismic Detection

**DOI:** 10.3390/s19173734

**Published:** 2019-08-29

**Authors:** Yao Wang, Nengyi Fu, Zhihong Fu, Xinglin Lu, Xian Liao, Haowen Wang, Shanqiang Qin

**Affiliations:** 1State Key Laboratory of Power Transmission Equipment and System Security and New Technology, Chongqing University, No. 174 Shazhengjie, Chongqing 400044, China; 2School of Electrical Engineering, Chongqing University, No. 174 Shazhengjie, Chongqing 400044, China; 3Department of Geophysics, Colorado School of Mines, Golden, CO 80401, USA; 4Chongqing Triloop Detection Co., Chongqing 402660, China; 5School of Robot Engineering, Yangtze Normal University, Chongqing 408100, China

**Keywords:** tunnel seismic detection, piezoelectric sensor, semi-automatic coupling, geophone, forward modeling

## Abstract

The tunnel seismic method allows for the detection of the geology in front of a tunnel face for the safety of tunnel construction. Conventional geophones have problems such as a narrow spectral width, low sensitivity, and poor coupling with the tunnel wall. To tackle issues above, we propose a semi-automatic coupling geophone equipped with a piezoelectric sensor with a spectral range of 10–5000 Hz and a sensitivity of 2.8 V/g. After the geophone was manually pushed into the borehole, it automatically coupled with the tunnel wall under the pressure of the springs within the device. A comparative experiment showed that the data spectrum acquired by the semi-automatic coupling geophone was much higher than that of the conventional geophone equipped with the same piezoelectric sensor. The seismic data were processed in combination with forward modeling. The imaging results also show that the data acquired by the semi-automatic coupling geophone were more in line with the actual geological conditions. In addition, the semi-automatic coupling geophone’s installation requires a lower amount of time and cost. In summary, the semi-automatic coupling geophone is able to efficiently acquire seismic data with high fidelity, which can provide a reference for tunnel construction safety.

## 1. Introduction

Along with transportation, water resource transportation, municipal pipelines, and mining projects, China’s tunnel and underground space engineering is developing at an unprecedented rate [1,2]. However, the construction of tunnels is accompanied by numerous geological disasters, including collapse, rock-burst, gushing water, and gushing mud. Therefore, it is important to pay more attention to construction safety in tunnel engineering [3,4,5,6,7]. It is necessary to understand the geology in front of the tunnel face prior to excavation. Tunnel seismic methods are extensively used for the detection of the geology in front of the tunnel face due to their long detection range and high resolution [8,9,10,11,12].

The theory of seismic detection is based on the fact that rocks noticeably differ in impedance value from type to type. Tunnel seismic detection allows one to measure and analyze reflected waves caused by the impedance contrast due to lithological differences. In engineering applications, the tunnel seismic method is often conducted by using explosives and geophones installed either on the tunnel face or on the tunnel side walls. The geology in front of the tunnel face can be obtained by analyzing the reflected wave acquired by geophones [13].

Seismic detection is among the most effective means of mapping subsurface structures and properties in the oil industry [14]. Seismic methods are able to detect large-scale geological bodies in the oil industry. The spectrum of oil seismic signals can be up to 200 Hz [15], and the detection depth can be up to 10,000 m. Oil seismic exploration mainly uses explosives of up to tens of kilograms as the seismic source [16,17]. The electromagnetic coiled geophone is a proven technology that provided the industry with rugged, cheap, and self-powered geophones [18]. However, this type of geophone is not suitable for the tunnel seismic method. The tunnel seismic method applies 20–100 g of explosives inside of boreholes drilled along the tunnel rock wall to achieve an illumination range of 100–200 m in front of the tunnel face. Because only a low amount of explosive is deployed in the rock’s borehole and the offsets (the distances between all possible combinations of shots and geophones) are all equally short in the tunnel, the spectrum of the seismic signal is broad. The bandwidth of the electromagnetic coiled geophone is too narrow to meet the demand. In addition, since the tunnel seismic detection range is short and the seismic wave propagates in the whole space, the seismic wave field is complicated, and the signal energy in the tunnel, such as that of direct waves, Rayleigh waves, and acoustic waves, dominates over that of the reflected wave resulting from the geological body in front of the tunnel face. Only the reflected wave of the geological body in front of the tunnel face provides the information that meets our interests. In addition to low bandwidth, due to the low sensitivity of the electromagnetic coiled geophone, it is difficult to acquire accurate reflected signals. Piezoelectric sensors have significant advantages over electromagnetic coiled sensors in terms of bandwidth and sensitivity, and they are used in a wide range of applications in areas of condition and safety monitoring for various industrial facilities [19,20,21]. In the oil industry, piezoelectric sensors are mainly used for high-precision geological structures and lithology exploration, where high-frequency seismic signals can significantly improve resolution [22,23,24]. In short-distance geological exploration, the advantages of piezoelectric sensors are more obvious. In cross-well seismic detection, a piezoelectric sensor is able to acquire broadband seismic signals for high-resolution tomography [25,26]. In ultra-shallow seismic reflection surveying, the high-frequency seismic data (up to 2000 Hz) acquired by piezoelectric sensors can accurately result in an image of the fracture zone, whereas electromagnetic coiled sensors fail to acquire a sufficient amount of reflection seismic data due to their narrow bandwidth [27]. The spectrum of channel waves can be up to 570 Hz in coalmine seismic detection; thus, piezoelectric geophones with a wide frequency band and stable sensitivity are essential for seismic data acquisition in coal seam channel wave exploration [28].

To accurately record ground motion for seismic detection, geophones must be firmly coupled to the ground [29,30]. Note that there exists a major difference in coupling between oil seismic and tunnel seismic methods. In oil seismic detection, the tails of the geophones are buried vertically into the surface soil to achieve better coupling. Coupling is noticeably more challenging in the tunnel simply because tunnels essentially consist of hard rocks. There exist three coupling methods which are mainly used in the industry. The most widely used method is known as the casting method [8]. Firstly, steel pipes are installed via proper methods into the boreholes drilled along tunnel walls, and the geophone is then installed into the steel pipes. Via this method, acoustic waves and Rayleigh waves can be considerably reduced, and the influence of the tunnel surface’s low-speed circle caused by excavation can be tremendously mitigated. Nevertheless, this method has a few shortcomings. Firstly, the geophones are designed in such a way that they can move freely back and forth within the steel pipes; as a result, the coupling is not sufficient. Secondly, this method is time-consuming, where the installation time for a single geophone can be up to half an hour, and it has a relatively high cost, where a single data sampling using this method costs about 700 United States dollars (USD). The second method is the clay coupling method, which is widely implemented in China [31]. In this method, a 2-m-deep hole in the tunnel wall is firstly drilled, and clay is then applied before pushing the geophone into the borehole to reinforce the coupling, since the gaps between the geophone and borehole are filled. The second method is relatively cheaper compared to the first one, yet it is unable to have the noise reduction advantages of the first method. The third method is a direct coupling method; in this method, geophones are directly attached to the tunnel wall with quick-drying cement [32]. However, this method can potentially capture a considerable amount of noise, such as acoustic waves and Rayleigh waves, as well as the influence of the tunnel surface’s low-speed circle caused by excavation.

Therefore, it is in our favor to develop an innovative design of a tunnel geophone that ensures the high-efficiency acquisition of high-quality seismic data. We installed a wide-band, high-sensitivity piezoelectric sensor in the geophone and designed a semi-automatic coupling mechanism. The new geophone was compared with a conventional geophone in a tunnel, and the results indicate that the new geophone is superior to the conventional one.

## 2. Materials and Methods

### 2.1. The Semi-Automatic Coupling Geophone

#### 2.1.1. Piezoelectric Sensors

Previous studies showed that high-frequency seismic waves excited in rocks can be up to 2 kHz [27]; therefore, a sensor with a spectrum range of 10-5kHzis able to capture such a signal. However, we still have to adjust the sensitivities of sensors accordingly prior to tunnel seismic detection, because the sensitivity and range of a sensor are inversely related. A sensitivity of 2.8 V/g is a suitable value for the tunnel seismic method. The specifications of the three-component piezoelectric sensor are shown in Table 1, where X, Y, and Z represent the X-component, the Y-component, and the Z-component, respectively. The amplitude responses are shown in Figure 1. The sensor was tested using two vibration platforms: Figure 1a shows the low-frequency signals, and Figure 1b shows the medium- and high-frequency signals.

In order to understand the advantages of the piezoelectric sensor, a comparison test between the piezoelectric sensor and the electromagnetic coiled sensor in a tunnel was conducted. Since the electromagnetic coiled sensor involves a single component, the three sensors were combined together with an iron shell (Figure 2), and the three sensors were placed perpendicular to each other to ensure that data were acquired in different directions. Two holes with distances of 30 m were drilled in the tunnel wall; then, 30 g of explosive was shot in one hole, and the piezoelectric sensor and the electromagnetic coiled sensor were placed in the other hole to acquire the seismic signal. Seismic data were recorded by a TETSP-2 tunnel seismometer (Chongqing Triloop Detection Co., Chongqing, China).

#### 2.1.2. The Semi-Automatic Coupling Geophone

To ensure high-quality seismic data acquisition, a 2-m-deep hole was firstly drilled for geophone installation. Afterward, we applied the new semi-automatic coupling geophone which consisted of three parts: a geophone, a metal rod, and a handle (Figure 3). The geophone used a piezoelectric sensor, a wheel, and a junction box (Figure 3a). The wheel was designed and attached for the purpose of coupling; more specifically, the center of the wheel was attached with two hard springs so that the wheels could rotate and move in a vertical manner (Figure 3b and Figure 4). Once the device was pushed into the borehole, the wheel could start to roll along the surface of the sidewall of the borehole, aiding the installation of the geophone tremendously (Figure 5). In a nutshell, this method guaranteed a fair coupling and easy removal. After the device was pushed into the borehole, a wedge-shaped stone block was applied to clamp the handle to reduce vibration. Afterward, the hole was sealed with clay to reduce acoustic waves. Nevertheless, new issues with this approach could arise; rock slag could start to accumulate and sink into the cavity below the wheel, thereby blocking the wheel. To confront with this problem, a sleeve under the wheel was designed to remove the accumulating rock slag. After each use, the rock slag could be discharged from the cavities by washing.

### 2.2. Field Comparison Experiment

In order to study the coupling effects of the purposed semi-automatic coupling geophone, we designed a conventional geophone using the same piezoelectric sensor and conducted a comparative experiment in a tunnel in Chongqing, China. The surrounding rock of the tunnel was tight sandstone with good integrity. Twenty-six horizontal holes were drilled in the tunnel wall, including 24 shot holes and two receiving holes (Figure 6). This linear geometry was designed in the tunnel to detect the geology in front of the tunnel face [8]. The semi-automatic coupling geophone was pushed directly into one of the receiving holes, a wedge-shaped stone block was applied to clamp the handle, and the hole is sealed with clay, which took about one minute. Using the clay coupling method to install a conventional geophone in another adjacent receiving hole, we also designed auxiliary equipment to install the conventional geophone (Figure 7a). Firstly, about 1 kg of clay with moderate water content was prepared. If the water content of the clay deviates too much from the specification, the coupling effect is significantly reduced. The shovel was connected to the push rod, and the clay was then placed into the shovel and pushed into the bottom of the hole. After removing the push rod from the hole, the shovel was replaced with a fork. The fork was pushed into the two holes at the end of the geophone (Figure 7b), and the geophone was then pushed gently into the borehole while ensuring that the direction was straight. After inserting the geophone into the clay, the push rod was gently tapped with a heavy hammer. The geophone was fully coupled to the borehole. The push rod was then gently removed from the hole, the geophone was left in the borehole, and the hole was finally sealed with clay. The entire process took 26 min. After installing the two geophones, the explosives were excited sequentially in the 24 holes. Seismic data were recorded by a TETSP-2 tunnel seismometer (Chongqing Triloop Detection Co., Chongqing, China). Since the tunnel occupied the whole space, and the boundary conditions were complex, there are many other waves in addition to direct waves and acoustic waves. These waves were coincident and difficult to distinguish. Therefore, forward modeling was needed to understand the characteristics of various seismic waves in the tunnel, and the data processing was then performed to extract the reflected waves in front of the tunnel face, so as to realize the imaging of the geological body in front of the tunnel face.

### 2.3. Seismic Data Processing Combined with Forward Modeling

#### 2.3.1. Numerical Calculation

In order to study the characteristics of the reflected wave in front of the tunnel face, we designed a simple two-dimensional geological model (Figure 8), which consisted of a tunnel and three media. There were 24 shots and one receiver in the tunnel seismic geometry.

Based on the elastic wave equation, we numerically simulated the above model. The two-dimensional wave equation in an elastic, isotropic, and heterogeneous medium is described as follows:(1)$$\left\{ \begin{array}{l} \rho \frac{\partial v_{x}}{\partial t}=\frac{\partial \tau_{xx}}{\partial x}+\frac{\partial \tau_{xz}}{\partial z}\\ \rho \frac{\partial v_{z}}{\partial t}=\frac{\partial \tau_{xz}}{\partial x}+\frac{\partial \tau_{zz}}{\partial z}\\ \frac{\partial \tau_{xx}}{\partial t}=(\lambda +2\mu )\frac{\partial v_{x}}{\partial x}+\lambda \frac{\partial v_{z}}{\partial z}\\ \frac{\partial \tau_{zz}}{\partial t}=\lambda \frac{\partial v_{x}}{\partial x}+(\lambda +2\mu )\frac{\partial v_{z}}{\partial z}\\ \frac{\partial \tau_{xz}}{\partial t}=\mu (\frac{\partial v_{z}}{\partial x}+\frac{\partial v_{x}}{\partial z}) \end{array}\right.,$$
where *v_z_* and *v_x_* are the velocities of the medium, *τ_xx_* and *τ**_zz_* are normal stresses, *τ**_xz_* is the shear stress, *ρ* is the medium density, *λ* and *μ* are the Lame constant, and t is the time. The finite difference method was used to perform numerical calculations [34,35]. The space step of the model was 0.4 m. A 500-Hz Ricker wavelet was used as the source, and the boundary conditions were selected as the perfectly matched layer (PML) boundaries. Due to the particularity of the tunnel conditions, free boundary conditions were set on the tunnel wall and the tunnel face, respectively, and the stress was set to zero [36]. The above conditions were used for the programming calculation.

#### 2.3.2. Field Data Processing

Tunnel seismic data processing included three parts: pre-processing, processing, and interpretation (Figure 9). Pre-processing involved importing data into a computer and doing a simple data preview and correction. The purpose of data processing was to extract the reflected waves of the geological body in front of the tunnel face and use it to image the geological body. The arrival time was determined firstly to calculate the direct wave velocity by the least square method. The root mean square (RMS) was used to equalize the amplitude of the seismic data. The automatic gain control (AGC) was used to obtain the bottom data amplitude. The band-pass filter was used to filter out the surface waves, acoustic waves, and other noise in the tunnel. The F–K (where F is frequency, and K is wave number) filter could be applied to transform the seismic data from the time domain to the frequency wave-number domain [37]. In the frequency wave-number domain, the seismic waves reflecting from the tunnel face direction and the opposite direction are distributed in different quadrants due to the opposite of the apparent velocity; thus, the reflected waves in front of the tunnel face can be easily extracted. The polarization filter was used to extract P-waves [38]. Since it is difficult to obtain an accurate S-wave velocity in real tunnel seismic data processing, we extracted three components of P-waves for imaging. We used the equi-travel time plane migration method to process the polarization filtered seismic data [39,40]. Since the aperture in the tunnel was too small, the migration results were curved. These curves did not represent the true form of the geological interface, but the homogeneity of the geological conditions could be inferred by the amplitude of these curves. In the interpretation process, the instantaneous amplitude attribute was used to better distinguish the energy of the profile. The stronger the instantaneous amplitude attribute was, the worse the geological conditions were. Finally, a comprehensive analysis of the geological conditions in the front of the tunnel face was combined with the instantaneous amplitude attribute and the existing geological data.

## 3. Results and Discussion

### 3.1. Comparison Test between the Piezoelectric Sensor and the Electromagnetic Coiled Sensor

We found that the same seismic signal acquired by the two sensors had a large difference in the waveform obtained (Figure 10). The starting point of the data acquired by the piezoelectric sensor was easy to identify, and the data acquired by the electromagnetic coiled sensor showed evident waveform oscillation. The spectral range of the data acquired by the piezoelectric sensor was much broader than that of the electromagnetic coiled sensor (Figure 11). When the high-frequency portion was acquired by the piezoelectric sensor, it gradually attenuated until 1500 Hz; however, when it was acquired by the electromagnetic coiled sensor, it sharply attenuated until 500 Hz. The low-frequency portion acquired by the piezoelectric sensor was more abundant than that of the electromagnetic coiled sensor.

These oscillation signals acquired by the electromagnetic coiled sensor were not real vibration signals. Because the spectrum of the tunnel seismic signal was wider than the spectral range of the electromagnetic coiled sensor, these waveform oscillations were caused by the band-pass filtering of the original signal by the sensor. Therefore, the spectrum curve of the data acquired by the electromagnetic coiled sensor looks steep (Figure 11). This shows that the seismic data were distorted and did not reflect the true propagation of the seismic waves in the tunnel rock. In addition, the first arrival wave picking was inaccurate due to the oscillations, resulting in inaccurate subsequent F–K filtering and migration results.

The range and sensitivity of the piezoelectric sensor are inversely related. The seismic wave energy was strong due to the small offset in the test, and the normal seismic waveforms indicated that energy was within the sensor’s range. Hence, it was reasonable to choose a sensor with such high sensitivity. The spectral curve of the seismic data acquired by the piezoelectric sensor was slowly attenuated at 1500 Hz, indicating that the data spectrum range was within the bandwidth. This shows that the piezoelectric sensor we chose dominated over the electromagnetic coiled sensor and was, therefore, a reasonable choice in the acquisition of tunnel seismic signals.

### 3.2. Field Comparison Experiment

Seismic data acquired by the two geophones were recorded by a TETSP-2 tunnel seismograph (Figure 12 and Figure 13). The consistency of the tri-component data of both geophones was good, and the direct waves and the acoustic waves could be clearly found, indicating that the sensor was reliable. Through the spectrum analysis of seismic data (Figure 14), we found that the spectrum of the X-, Y-, and Z-components of the semi-automatic coupling geophone could be up to 2000, 1000, and 2000 Hz, respectively, while those of the conventional geophone were 600, 600, and 500 Hz, respectively. The spectrum of the X-component and the Z-component data of the semi-automatic coupling geophone was significantly higher than that of the conventional geophone, and the Y-component was relatively higher. Previous studies explained whether coupling being tight has a huge impact on data [41,42]. Essentially, clay is a soft medium and it is impossible to form a hard coupling. The installation of geophone in 2-m-deep boreholes further increased the uncertainty of coupling. The high-frequency portion acquired by the piezoelectric sensor was gradually attenuated, and the waveform was normal. This shows that the semi-automatic coupling geophone acquired seismic data with higher fidelity. Previous studies showed that the spectrum of the tunnel seismic method is within 1 kHz [12,33], which is related to the lack of coupling. Of course, it is also related to the geological conditions of the tunnel. This study proves that the spectrum of the seismic method in a tight sandstone tunnel can reach 2 kHz, and rich high-frequency data can improve the resolution of imaging.

In addition, the semi-automatic coupling geophone reduced installation time from 26 min to one min; therefore, cost and time are considerably reduced compared with conventional tunnel seismic detection. Since tunnel seismic detection is implemented in the tunnel construction gap, reducing the detection time can also make the construction more convenient. Finally, the semi-automatic coupling geophone installation does not require any auxiliary equipment, saving costs.

### 3.3. Forward Modeling

The forward modeling data are shown in Figure 15. Firstly, the direct wave and the Rayleigh wave could be directly found. The Rayleigh wave was generated in the tunnel wall and had evident dispersion characteristics. Since the direct wave and the Rayleigh wave propagated directly from the source to the geophone without any reflection, the energy was strong. It can be seen that the face P-wave reflected by the tunnel face, as well as the Rayleigh wave, partially coincided with the X-component data (Figure 15a). A face P-wave in the Y-component could not be found because of the weaker energy due to the P-wave’s polarization (Figure 15b). In theory, there should be a face PS (P-to-S) wave reflected by the tunnel face followed by the face P-wave [26]; however, the size of the face was too small, and the energy of the face PS wave was too weak to be found. The IP-wave (P wave reflected by interface I) could be found under the face P-wave. The IP-wave is an important wave for tunnel seismic detection. We can image interface I by jointly processing the IP-wave and the IPS wave (P-to-S wave reflected by interface I). Rayleigh waves arriving at the tunnel face were converted into a high-amplitude RS (Rayleigh-to-S) wave followed by an IP-wave [27]. The energy of the RS wave was too strong and completely coincided with the IPS wave and the IIP-wave (P wave reflected by interface II), causing the IPS wave and the IIP-waveto be unrecognizable; therefore, we marked their positions according to the wave travel time. Both of these waves are critical for imaging. Another important wave at the bottom of the time axis is the IIPS (P-to-S wave reflected by interface II) wave, which is the converted S-wave of interface II. Since the direct wave and the Rayleigh wave were too strong, the IIPS wave energy here was relatively weak and difficult to recognize, and the Y-component (Figure 15b) data were slightly more obvious than the X-component data (Figure 15a) due to the polarization direction. In order to study the reflected wave in front of the tunnel working face, there was no design reflection interface behind the tunnel face in the model. It is reasonable to speculate that there were still a large number of reflected waves from the geologic body behind the tunnel face in the actual tunnel seismic method. By analyzing the forward data, an evident law was discovered. The transmitted wave (including the direct wave and the Rayleigh wave) was tilted to the lower right, and the reflected wave was inclined to the upper right. The reflected waves that were tilted to the upper right needed to be extracted, and these reflected waves were used to image the geology in front of the tunnel face. Previous forward modeling mainly focused on the shape of the geological body in the front of the tunnel face, without considering the tunnel boundary [10,43]. This study found that the tunnel wall and the face of the face generate strong-amplitude surface waves and RS waves, which must be carefully filtered to obtain the reflected waves in the tunnel’s advanced direction.

### 3.4. Field Data Processing

Figure 16 and Figure 17 show the F–K-filtered seismic data acquired by the semi-automatic coupling geophone and the conventional geophone, respectively. Both events were clearly inclined to the upper right. According to our comparison with the forward model (Figure 15), these waves were reflected waves in front of the tunnel face. The events of the semi-automatic coupling geophone data (Figure 16) were finer than those of the conventional geophone data (Figure 17), which indicates that the resolution of the semi-automatic coupling data was superior and beneficial for imaging.

Figure 18 shows the migration profile of the P-wave acquired by the semi-automatic coupling geophone and the conventional geophone. Since the aperture in the tunnel was too small, the migration results were curved. These curves did not represent the true form of the geological interface, but the homogeneity of the geological conditions could be inferred by the amplitude of these curves. The instantaneous amplitude attribute was used to better distinguish the energy of the profile (Figure 19). Previous studies focused on imaging with different migration methods [8,9,12,32,39]. This study extracted instantaneous amplitude properties from the migrated data, clarifying the imaging of the heterogeneous geological body. The stronger the instantaneous amplitude attribute is, the worse the geological conditions are. For example, karsts, weak interlayers, fracture zones, and joint fissure development can exhibit a strong amplitude in the instantaneous amplitude property. The geological data must be closely integrated to improve the accuracy of detection. We found that the strong-amplitude section of the semi-automatic coupling data was at mileages of 760–782, 823–841, and 880–888, while the conventional geophone showed mileages of 768–788, 815–858, and 868–884 (Figure 19). The data acquired by the conventional geophone had a larger region of strong amplitudes, and this was related to the spectrum of the data. Because the coupling was not tight, the high-frequency data were lost, and the resolution was lowered. The geological data of the tunnel indicated that the surrounding rock conditions were good; thus, it was speculated that there was no geological hazard present within the entire detection section, and the abovementioned strong-amplitude section may have developed joint fissures. The semi-automatic coupling result had a small-amplitude segment, and the amplitude of the entire profile was relatively low; thus, it was more consistent with the actual geological data. The subsequent excavation also revealed this.

## 4. Conclusions

An innovative semi-automatic coupling piezoelectric geophone was proposed and designed here to facilitate tunnel seismic detection. In particular, a piezoelectric sensor was installed in our new geophone with a spectral width of 10–5000 Hz and a sensitivity of 2.8 V/g. The piezoelectric sensor and the electromagnetic coiled sensor were tested in the tunnel. By comparing the results, the seismic waveform acquired by the piezoelectric geophone was normal and the spectrum was noticeably wide, while the seismic data acquired by the electromagnetic geophone suffered from waveform distortion and had narrow spectrum bandwidth. This shows that the piezoelectric sensor dominates over the electromagnetic coiled in terms of data quality.

A semi-automatic coupling design was implemented to tackle the problem of coupling the geophone to the borehole of the tunnel wall. The design enabled a tight contact between the geophone and the tunnel wall, ensuring the acquisition of high-quality seismic data. Through field comparison experiments with the conventional geophones, the semi-automatic coupling design had several following advantages. Firstly, data acquired by the new geophone had higher fidelity. Secondly, the installation time was reduced from 26 min to one min, thereby considerably reducing the time cost, allowing workers to avoid the harsh conditions for tunnel seismic surveys in the construction stage. Finally, the new geophone did not require any auxiliary equipment or coupling agents. Considering that there are a considerable number of tunnels still in the construction stage in China, large-scale productions of the new geophones can help in mitigating the total cost of tunnel construction in the long run.

The numerical simulation of tunnel seismic detection showed that the seismic wave in the tunnel was much more complicated than the seismic wave generated via the oil seismic method. In general, direct waves and Rayleigh waves are generated within the tunnel wall, the RS wave is generated by the tunnel face, and the reflected P- and PS waves of the geological body are generated in front of the tunnel working face. These waves partially overlap; thus, they are difficult to identify. In a nutshell, a series of conclusions can be deduced and concluded from forward modeling. The energy of the Rayleigh wave and the RS wave was strong, and the Rayleigh wave showed obvious dispersion. The reflected wave of the geological body in front of the tunnel face was inclined to the upper right. The above conclusions can help us analyze tunnel seismic waves and are beneficial in processing the actual tunnel seismic data.

The tunnel seismic data processing flow was designed. The data, after being F–K filtered, could be used to easily find events inclined to the upper right. This shows that the reflected waves ahead of the tunnel working face were acquired. The events of the data acquired by the semi-automatic coupling geophone were finer than those of the conventional geophone, which is more advantageous for imaging. We used the instantaneous amplitude attribute to reflect the homogeneity of the medium in front of the tunnel face. The instantaneous amplitude attributes of these data had significantly strong amplitudes, and the mileage was similar. By comparison with geological data, it was found that the data acquired by the semi-automatic coupling geophone were more in line with the actual situation. This further indicates that the semi-automatic coupling geophone acquires high-fidelity tunnel seismic data, which can reinforce safety for tunnel construction.

Three-dimensional high-precision tunnel seismic detection is a subject of future interest. It may require the installation of 10 or more geophones in a tunnel to realize three-dimensional detection. The proposed semi-automatic coupling geophone has a short installation time and low cost, which can be used in three-dimensional tunnel seismic detection. More importantly, the geophone can acquire high-fidelity seismic signals in a tunnel, and the frequency can be up to 2000 Hz. In this study, simple processing and analysis were performed on the signal. High-resolution imaging studies should be performed on this signal in the future.

## Figures and Tables

**Figure 1 sensors-19-03734-f001:**
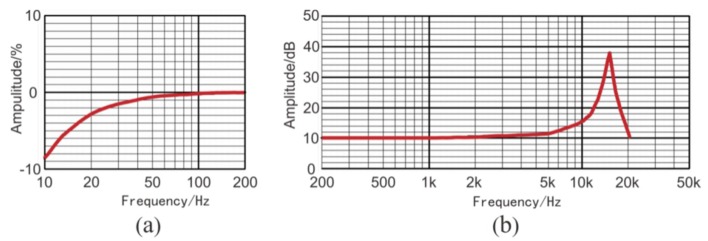
The amplitude responses of the three-component piezoelectric sensor: (**a**) low frequency; (**b**) medium and high frequency [33].

**Figure 2 sensors-19-03734-f002:**
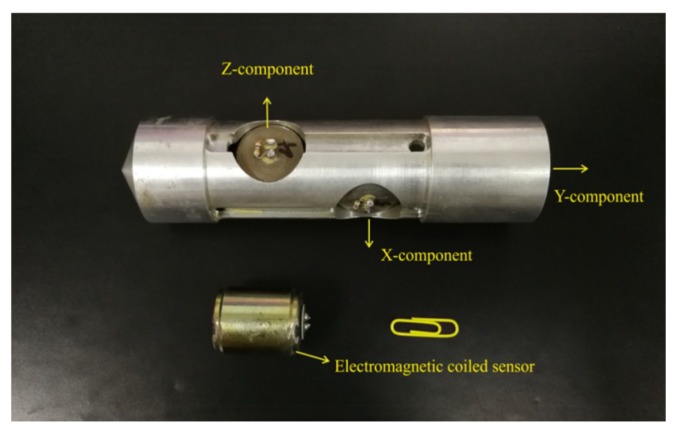
The tri-component electromagnetic coiled sensors. The natural frequency and sensitivity of the sensor were 60 Hz and 0.6 V/(m/s), respectively.

**Figure 3 sensors-19-03734-f003:**
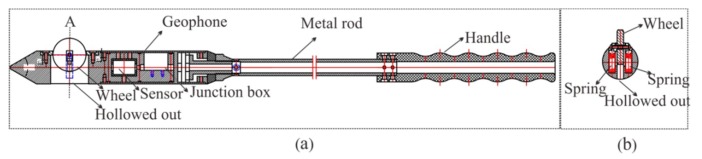
Cutaway view of the semi-automatic coupling geophone: (**a**) side view; (**b**) cross-sectional view of wheel [33].

**Figure 4 sensors-19-03734-f004:**
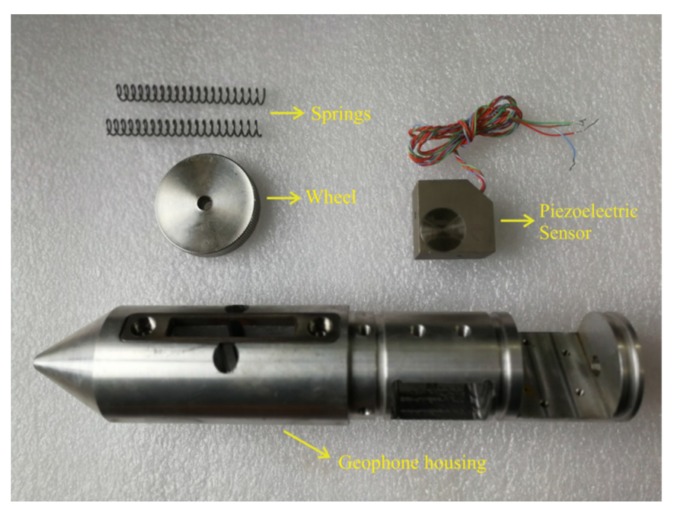
Internal structure diagram of the semi-automatic coupling geophone [33].

**Figure 5 sensors-19-03734-f005:**
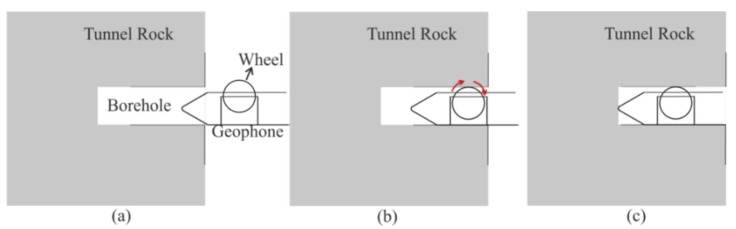
The semi-automatic coupling geophone installation process: (**a**) before, (**b**) during, and (**c**) after installation [33].

**Figure 6 sensors-19-03734-f006:**
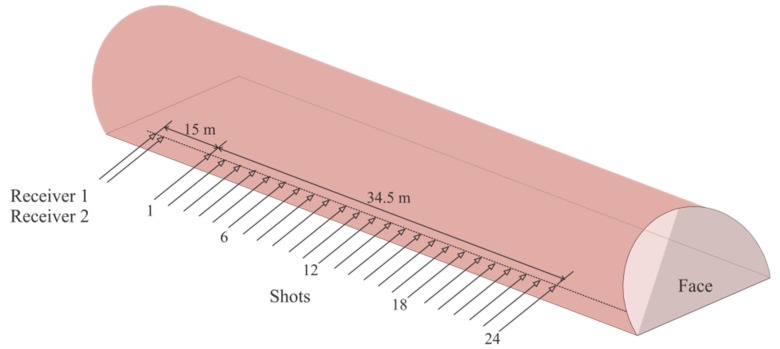
Tunnel seismic geometry, in which the minimum offset was about 30 m, and the borehole span was 2 m. Receiver 1 was the semi-automatic coupling geophone, and receiver 2 was the conventional geophone. The distance between the two geophones was about 0.5 m. For this test, 30 g of explosives were excited sequentially in the 24 holes.

**Figure 7 sensors-19-03734-f007:**
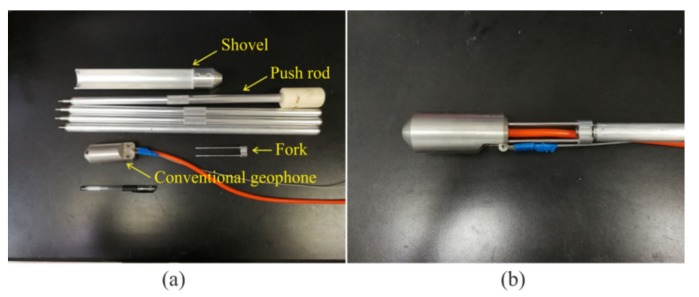
Picture of a conventional geophone and auxiliary equipment: (**a**) the auxiliary equipment consisted of four push rods, a shovel, and a fork. (**b**) Two small holes were designed in the tail of the geophone to insert the fork into the hole and push the geophone into the receiving hole.

**Figure 8 sensors-19-03734-f008:**
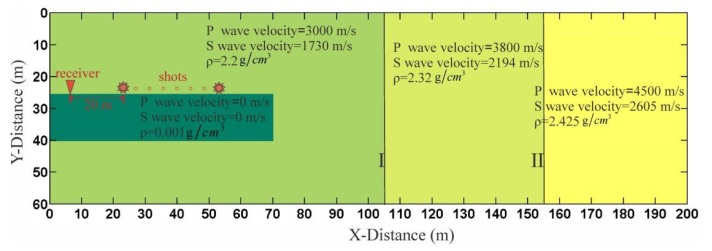
Sketch of the geological model. I and II are the two wave impedance interfaces, respectively. Different colors represent different media.

**Figure 9 sensors-19-03734-f009:**
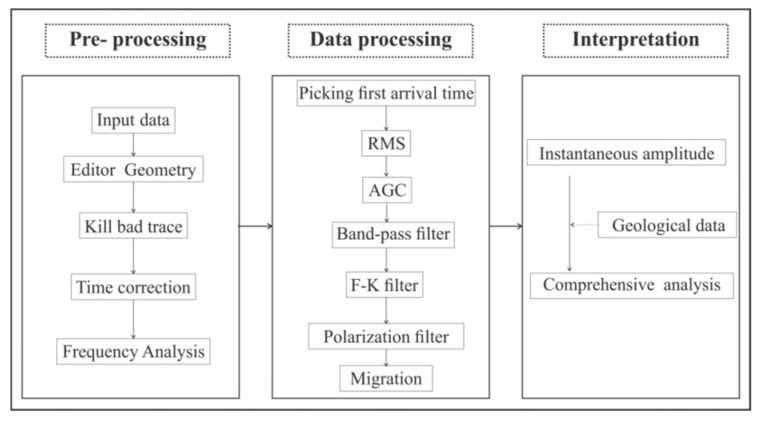
Tunnel seismic data processing flowchart.

**Figure 10 sensors-19-03734-f010:**
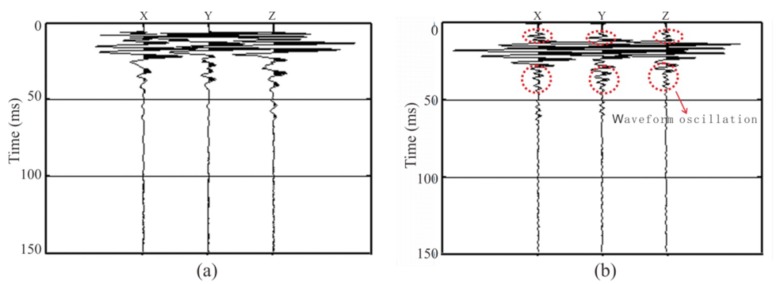
Spectral analysis of data acquired by the piezoelectric sensor and the electromagnetic coiled sensor: (**a**) piezoelectric sensor; (**b**) electromagnetic coiled sensor.

**Figure 11 sensors-19-03734-f011:**
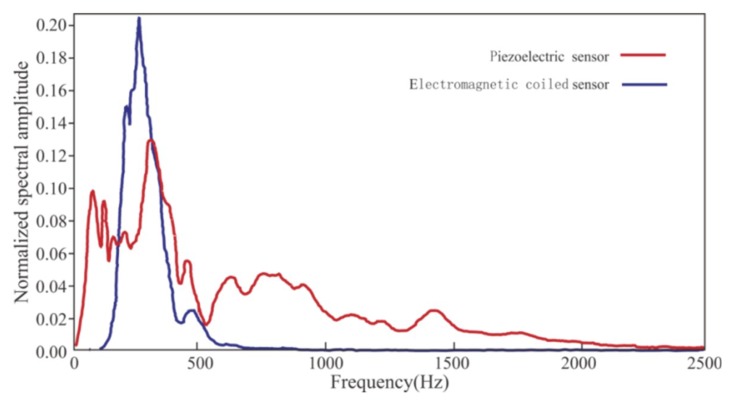
Spectral analysis of data acquired by the piezoelectric sensor and the electromagnetic coiled sensor.

**Figure 12 sensors-19-03734-f012:**
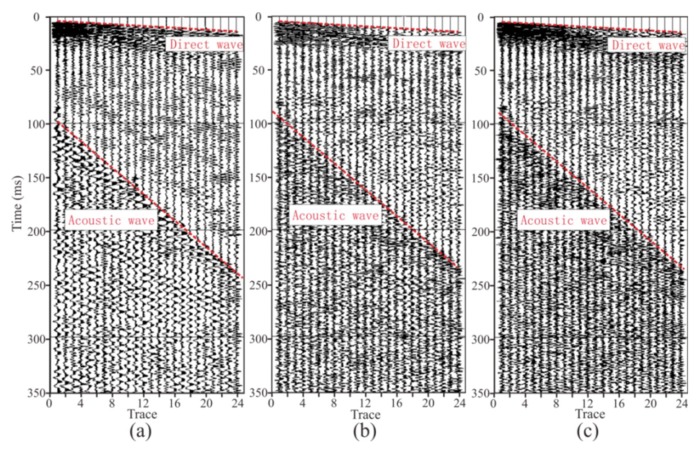
Seismic data acquired by the automatic coupling geophone. (**a**–**c**) are the X-component, the Y-component, and the Z-component data, respectively.

**Figure 13 sensors-19-03734-f013:**
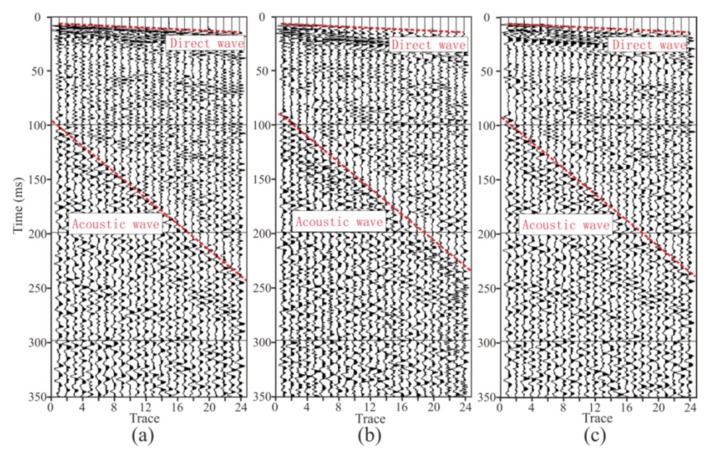
Seismic data acquired by the conventional geophone. (**a**–**c**) are the X-component, the Y-component, and the Z-component data, respectively.

**Figure 14 sensors-19-03734-f014:**
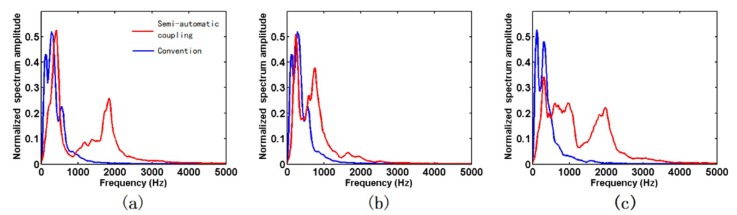
Spectrum comparison of component data of the two geophones, where (**a**–**c**) represent the spectra of the X-, Y-, and Z-components, respectively. The red and blue lines in the figure represent the data acquired by the semi-automatic coupling geophone and the conventional geophone, respectively.

**Figure 15 sensors-19-03734-f015:**
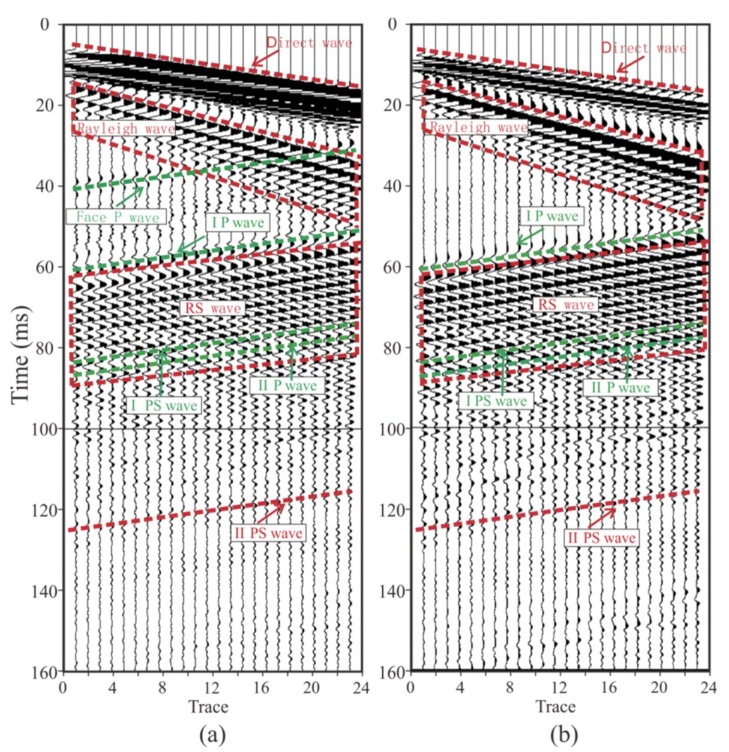
Forward modeling data: (**a**,**b**) are the X-component and the Y-component, respectively. The direct wave is a seismic wave that propagates directly from the borehole to the geophone; the Rayleigh wave is the wave generated by the tunnel wall; the RS wave is the converted S-wave of the tunnel face; the IP-wave and the IPS wave are the reflected P-wave and the converted S-wave of the interface I, respectively; the II P-wave and IIPS wave are the reflected P-wave and converted S-wave of interface II, respectively.

**Figure 16 sensors-19-03734-f016:**
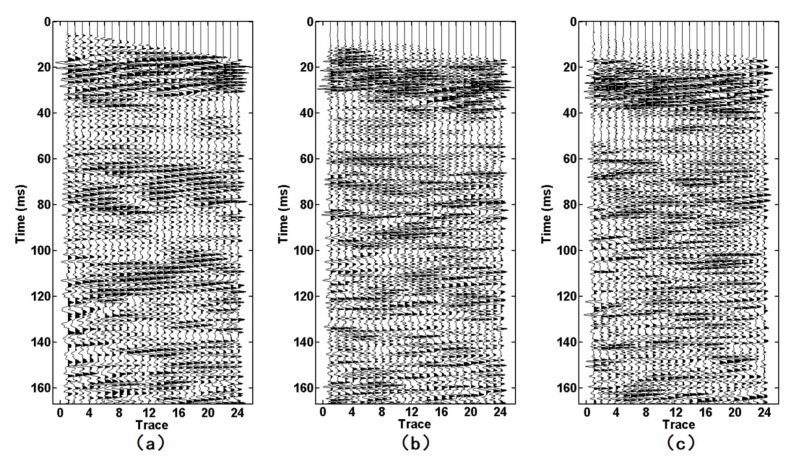
Reflected wave in front of the tunnel face: (**a**–**c**) are the X-component, Y-component, and Z-component, respectively.

**Figure 17 sensors-19-03734-f017:**
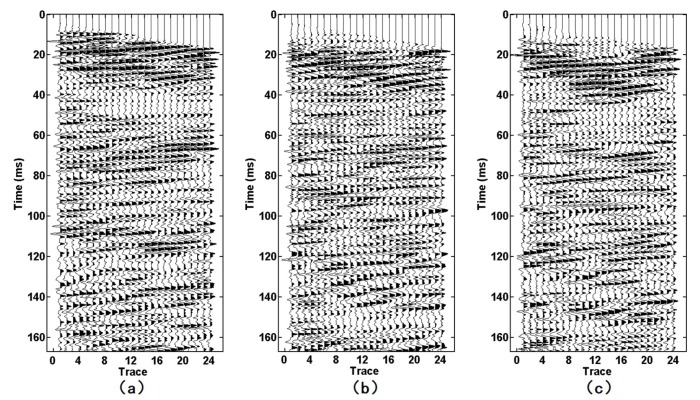
Reflected wave in front of the tunnel face: (**a**–**c**) are the X-component, Y-component, and Z-component, respectively.

**Figure 18 sensors-19-03734-f018:**
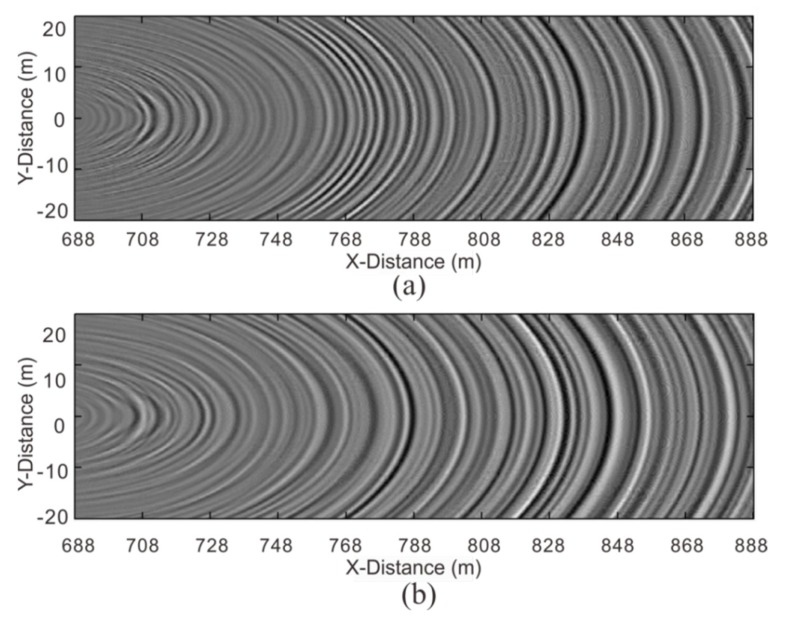
Migration profile of the P-wave. The results of superimposing the three component migration results into one profile: (**a**) the semi-automatic coupling geophone; (**b**) the conventional geophone.

**Figure 19 sensors-19-03734-f019:**
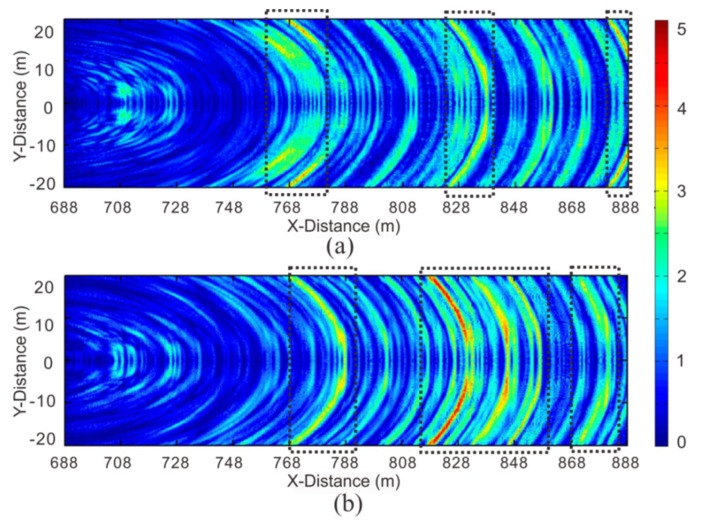
Instantaneous amplitude attribute profile: (**a**) the semi-automatic coupling geophone; (**b**) the conventional geophone.

**Table 1 sensors-19-03734-t001:** Specifications of the three-component piezoelectric sensor.

Description		Value
Sensitivity (mV/g)	X	2810
	Y	2830
	Z	2838
Full scale (g)		1.75
Frequency bandwidth (Hz)		10–5000
Resolution (g)		0.000006
Resonance frequency (Hz)		15,000

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
