# Peer review of "A Semi-Automatic Coupling Geophone for Tunnel Seismic Detection"

_sensors, 2019, doi:10.3390/s19173734_

Round 1

Reviewer 1 Report

This paper proposes the use of a semi-automatic coupling geophone equipped with a piezoelectric sensor as an alternative to overcome detection problems of conventional geophones. The subject is relevant and interesting. The experimental procedure is well detailed and the results are consistent. The reviewer recommends the manuscript for publication after a minor revision. Some issues should be considered as below:

- In the introduction section, lacks a paragraph devoted to present the background research of  the use of piezoelectric sensors for the detection of the geology.

- Lacks depth the dicussion of results of this paper. The authors separated the results of the discussions which left the text confusing. It is not usual to discuss the results in the same section of the conclusions. I suggest that the results and discussions be presented in section 3 and the conclusions in section 4.

- Another issue is that none publication was cited in the discussion of results. The authors need to better contextualize their results to literature  in order to emphasize  the contribution of the paper.

Author Response

Thank you very much for your review comments. Please find our response at the attachment.​

Reviewer 2 Report

The author introduces a semi-automatic coupling geophone equipped with a wide-band, high-sensitivity piezoelectric sensor. Compared with conventional geophones, there are many advantages, such as saving money and less time spent on equipment installation. At the same time, the equipment can efficiently acquire seismic data with high fidelity. Through a series of comparative tests, the applicability of the new geophone is verified and the test results are more accurate.

However, some presented questions have reduced the readability and quality of this paper. It’s the reviewer’s opinion that the paper should be substantially improved according to the following comments.

1. It’s noted that the manuscript needs discreet editing by someone with expertise in technical English. The English grammar and sentence structure should be revised further so that the goals and results of study are available to readers.

e.g.

p.1, line 34: 1st paragraph of Introduction part: “The construction of tunnels results in numerous challenges that include collapse, rock-burst, gushing water, and gushing mud…” should be changed into “However, the construction of tunnels is accompanied by numerous geological disasters, including collapse, rock-burst, gushing water, and gushing mud. Therefore, it is important to paid more attention to the construction safety of tunnel engineering”

p.14, line344: 1st paragraph of Discussion and Conclusions part: “The comparison results show that the data acquired by the electromagnetic coiled sensor show evident waveform oscillation…” should be changed into “By comparing the results, the data acquired by the electromagnetic coiled sensor shows evident waveform oscillations”

Please modify cautiously as the two examples above

2. In the introduction of the manuscript, some of the predecessors' research results are not enough, such as literature on equipment improvement and feasibility of research methods, please complete the addition.

3. p.3, line 97: please state the reason why the tri-component piezoelectric sensor with a spectral range of 10–5000 Hz was selected? Are there any preparations before selecting the sensor in the early stage?

4. p.9, line 247: The sentence that “we found that the spectrum of the X-, Y-, and Z-components of the semi-automatic coupling geophone can be as high as about 2000, 1000, and 2000 Hz, respectively, while those of the conventional geophone are 600, 600, and 500 Hz, respectively.is mentioned in the paper. Please explain in detail why the frequency between the two results varies greatly? Can it be understood as requiring higher fidelity?   

5. p.10, line 264: Please indicate the meaning of the red and blue lines in the legend in Figure 14.

6. In the section 4, Discussion and Conclusions of the manuscript are too long. It is recommended that the author put the detailed discussion in the section 3. The conclusions should be further streamlined and summarized.

7. Some format problems require the author to carefully proofread. For example, p.16, line 437: “Geophysics 1996” should be changed to “Geophysics 1996”.

8. The references should be completed, where many papers on the subject should be referred and acknowledged in the paper, such as

Numerical study on the seismic response of the underground subway station- surrounding soil mass-ground adjacent building system. Frontiers of Structural & Civil Engineering, 2017, 11(4): 424-435.

Geosynthetics used to stabilize vegetated surfaces for environmental sustainability in civil engineering. Frontiers of Structural & Civil Engineering, 2017, 11(4): 56-65.

Response characteristics of an atrium subway station subjected to bidirectional ground shaking. Soil Dynamics and Earthquake Engineering, 2019, 125: 105737.

Shaking table tests on discrepant responses of shaft-tunnel junction in soft soil under transverse excitations. Soil Dynamics and Earthquake Engineering, 2019, 120: 345-359.

An efficient stochastic dynamic analysis of soil media using radial basis function artificial neural network. Frontiers of Structural & Civil Engineering, 2017, 11(4): 470-479.

Author Response

(The authors gave the same response as above.)

Round 2

Reviewer 2 Report

The authors have substantially improved the paper and now it seems to be acceptable as it is.